# A Study on an Easy-Plane FeSi_3.5_ Composite with High Permeability and Ultra-Low Loss at the MHz Frequency Band

**DOI:** 10.3390/ma16145133

**Published:** 2023-07-21

**Authors:** Peng Wu, Shengyu Yang, Yuandong Huang, Guowu Wang, Jinghao Cui, Liang Qiao, Tao Wang, Fashen Li

**Affiliations:** Key Laboratory for Magnetism and Magnetic Materials of Ministry of Education, Institute of Applied Magnetism, Lanzhou University, Lanzhou 730030, China; wup21@lzu.edu.cn (P.W.); yangshy21@lzu.edu.cn (S.Y.); huangyd20@lzu.edu.cn (Y.H.); wanggw16@lzu.edu.cn (G.W.); cuijh21@lzu.edu.cn (J.C.); wtao@lzu.edu.cn (T.W.)

**Keywords:** easy-plane soft magnetic composites, MHz frequency band, ultra-low power loss, excess loss

## Abstract

An easy-plane FeSi_3.5_ composite with excellent magnetic properties and loss properties at MHz were proposed. The easy-plane FeSi_3.5_ composite has ultra-low loss at 10 MHz and 4 mT, about 372.88 kW/m^3^. In order to explore the reason that the $P_{cv}$ of easy-plane FeSi_3.5_ composite is ultra-low, a none easy-plane FeSi_3.5_ composite, without easy-plane processing as a control group, measured the microstructure, and the magnetic and loss properties. We first found that the real reason why magnetic materials do not work properly at MHz due to overheat is dramatical increase of the excess loss and the easy-plane composite can greatly re-duce the excess loss by loss measurement and separation. The total loss of none easy-plane FeSi_3.5_ composite is much higher than that of easy-plane FeSi_3.5_ composite, where the excess loss is a major part in the total loss and even over 80% in the none easy-plane FeSi_3.5_ composite. The easy-plane FeSi_3.5_ composite can greatly reduce the total loss compared to the none easy-plane FeSi_3.5_ composite, from 2785.8 kW/m^3^ to 500.42 kW/m^3^ (3 MHz, 8 mT), with the main reduction being the excess loss, from 2435.2 kW/m^3^ to 204.93 kW/m^3^ (3 MHz, 8 mT), reduced by 91.58%. Furthermore, the easy-plane FeSi_3.5_ composite also has excellent magnetic properties, high permeability and ferromagnetic resonance frequencies. This makes the easy-plane FeSi_3.5_ composite become an excellent soft magnetic composite and it is possible for magnetic devices to operate properly at higher frequencies, especially at the MHz band and above.

## 1. Introduction

With the development of electrical and electronic technology, soft magnetic materials are used as core components in daily life. Their magnetic properties and loss properties directly affect the power density and conversion efficiency [1]. In the further development of electrical and electronic devices, magnetic devices (inductor, power converters, etc.) are required to develop in the direction of miniaturization and high efficiency [2,3,4,5]. In the past, soft magnetic materials do not need to operate at higher frequencies due to the low operating frequency of power semiconductors. But now, the appearance of the third-generation wide bandgap semiconductors (WBG) has provided a high-frequency scenario to soft magnetic materials [6]. The conversion efficiency $P_{th}$ and size of soft magnetic materials in power devices (for example inductance, transformer, etc.) can be optimized because WBG increases the operating frequency, which can be described:(1)$$\begin{array}{c} P_{th}=CfB_{m}A_{e}W_{d} \end{array}$$
where C is the conversion efficiency coefficient, f is the operating frequency, $B_{m}$ is the maximum magnetic flux density, $A_{e}$ is the effective sectional area, and $W_{d}$ is the number of turns. However, soft magnetic materials have developed a problem, which makes them unable to work properly due to overheating at higher frequencies when WBG increases the operating frequency above MHz. Especially with the commercialization of SiC and GaN, the problem that soft magnetic materials cannot work properly at higher frequencies becomes more serious. A review, published in 2018 in *Science*, noted that none of the soft magnetic materials available today can cope with this daunting challenge. However, it also pointed out that soft magnetic composites (SMCs) have the potential to solve this problem [6]. 

Traditional SMCs are often used in electric motors, transformers, inductors and sensors due to high permeability, high-saturation magnetization and high-ferromagnetic resonance frequency [7,8,9,10]. When the devices try to work at higher frequencies, SMCs also have the same problem, which makes them unable to work properly due to overheating, resulting in a huge loss [6,11]. This overheating was thought to be caused by the eddy current loss at higher frequencies in the previous research [11,12,13,14]. Addressing the sharp increase in the loss at higher frequencies will provide a means of dealing with WBG and making soft magnetic materials work effectively.

The easy-plane soft magnetic materials are materials with a special magnetic configuration, where easy magnetization axes distributed in parallel planes with the same type enables the formation of the easy magnetization plane. According to the physical origins for the formation of the easy magnetization plane, easy-plane soft magnetic materials are divided into the magnetocrystalline-easy-plane soft magnetic materials (the easy magnetization axes are determined by the minimal of the magnetocrystalline anisotropy energy) and the magnetostatic-easy-plane soft magnetic materials (the easy magnetization axes are determined by the minimal of the magnetostatic energy). In the easy magnetization plane, the magnetization of easy-plane soft magnetic material is more easily saturated, remanence and coercivity of easy-plane soft magnetic materials are lower and the easy-plane soft magnetic materials has higher permeability and ferromagnetic resonance frequencies. When prepared as composites, the easy-plane soft magnetic composites maintain the excellent properties of the easy-plane soft magnetic materials, and also have the advantages of the composites. In the field of wave absorption, excellent results have been achieved due to the excellent performance of easy-plane soft magnetic composites [15,16,17,18]. Therefore, the easy-plane soft magnetic composites have the potential to address challenges posed by energy problem and WBG.

In the up-to-date published SMCs, measurement frequency of most studies are 1 MHz and below, such as FeSiCr (μ = 37, 560 kW/m^3^ at 1 MHz, 20 mT and μ = 47.5, 7086 kW/m^3^ at 1 MHz, 50 mT) [19,20], Fe_73_Si_6_B_10_P_5_C_3_Mo_3_ (μ = 34.63, ~500 kW/m^3^ at 1 MHz, 20 mT) [21], flake pure iron (μ = 67, 306.6 kW/m^3^ at 1 MHz, 3 mT and μ = 10, 3.285 kW/m^3^ at 3.5 MHz, 0.1 mT) [22], etc. There are almost no studies with high frequency and high maximum magnetic flux density. However, in this paper, we measured for the total loss $P_{cv}$ of the easy-plane FeSi_3.5_ composite at 10 MHz and 4 mT, and we found that the easy-plane FeSi_3.5_ composite has an ultra-low loss. The $P_{cv}$ of the easy-plane FeSi_3.5_ composite and none easy-plane FeSi_3.5_ composite was measured and analyzed at 3 MHz and 8 mT for this study; the reason for their ultra-low loss. We first found that the real reason why magnetic materials do not work properly at higher frequencies is due to overheating, which shows a dramatical increase in the excess loss of $P_{exc}$, and the easy-plane FeSi_3.5_ composite can greatly reduce the total loss of $P_{cv}$, and the main part of the reduction is the $P_{exc}$. The easy-plane FeSi_3.5_ composite has high permeability, high-ferromagnetic resonance frequencies and a lower loss compared with the none easy-plane FeSi_3.5_ composite by measurement and separation of loss. It is found that the sharp increase in the excess loss combined with frequency far exceeded the eddy current loss, and the excess loss becomes the most main part of total loss. The easy-plane FeSi_3.5_ composite not only has higher permeability and higher ferromagnetic resonance frequencies, but can also effectively reduce the excess loss. Thus, the easy-plane FeSi_3.5_ composite has the potential to deal with the overheating of soft magnetic materials due to excessive loss at higher frequencies, and is effectively utilized for devices at higher frequencies, especially at MHz and above.

## 2. Experiment

FeSi_3.5_ are the tradition spherical particles, obtained by purchasing. The tradition spherical FeSi_3.5_ particles are easy-plane processed to obtain the easy-plane FeSi_3.5_ particles, and the tradition spherical FeSi_3.5_ particles are named the none easy-plane FeSi_3.5_ particles. The easy-plane FeSi_3.5_ particles were compounded with polyurethane (PU) to prepare a polyurethane-based composite with 60 vol% (optimal volume fraction obtained after extensive experiments), and oriented in a rotating magnetic field (1 T) for 10 min. After the composite was heated to 90 °C, it was pressed into a ring-shaped composite with an outer diameter of 7 mm, an inner diameter of 3.04 mm and a thickness of 1 mm. And the none easy-plane FeSi_3.5_ composite was prepared using the same method.

The morphology of both particles and the composites were characterized by SEM (Apreo S, Thermo Fisher Scientific, Waltham, MA, USA). The vibrating sample magnetometer (VSM) (Microsence EV9, MicroSense, Lowell, MA, USA) was used to measure the hysteresis loop, and the degree of the plane orientation of two composites were also obtained by VSM data. The complex permeability of the composite was measured by a precision impedance analyzer (Agilent E4991B, Agilent, Santa Clara, CA, USA) and vector network analyzer (Agilent E8363B, Agilent, Santa Clara, CA, USA) at 1 MHz~18 GHz. The power loss was measured by a B-H analyzer (SY-8218, Iwatsu, Tokyo, Japan).

## 3. Result

The $P_{cv}$ of the easy-plane FeSi_3.5_ composite was measured at 10 MHz and 4 mT. It is found that the easy-plane FeSi_3.5_ composite has an ultra-low loss of about 372.88 kW/m^3^. The result is rare in SMC. In order to explore the reason that the $P_{cv}$ of the easy-plane FeSi_3.5_ composite is ultra-low, the none easy-plane FeSi_3.5_ composite, without easy-plane processing as a control group, measured the microstructure, magnetic properties and loss properties.

### 3.1. Morphology and Microstructure

Figure 1 shows the morphology and microstructures of the none easy-plane FeSi_3.5_ particles (a,b) and the easy-plane FeSi_3.5_ particles (d,e), cross sections of the none easy-plane FeSi_3.5_ (c) and easy-plane FeSi_3.5_ composite (f), respectively. The thickness of the none easy-plane FeSi_3.5_ particles is 44.35 μm, and the thickness of the easy-plane FeSi_3.5_ particles is 1.14 μm, thus the latter is much smaller than the former. The microstructures of the none easy-plane FeSi_3.5_ composite are a homogenous mixture of none easy-plane FeSi_3.5_ particles and PU; the microstructure of the easy-plane composite is an ordered laminar structure because of the orientation process.

### 3.2. Static Magnetic Performance and Degree of the Plane Orientation

Figure 2 shows the static magnetic properties of FeSi_3.5_ particles (a), hysteresis loops and the degree of plane orientation (*DPO*) of the none easy-plane FeSi_3.5_ composite (b) and easy-plane FeSi_3.5_ composite (c). The saturation magnetization ($M_{s}$) and the coercivity ($H_{c}$) of original FeSi_3.5_ is 184 emu/g and 3.32 Oe. The degree of plane orientation was characterized by measuring hysteresis loops of the in-plane and out-of-plane direction. The *DPO* can be expressed by Equation (2) [23]:(2)$$\begin{array}{c} DPO\left(\%\right)=\frac{M_{s}-M_{r,\;z-axis}}{M_{s}}\times 100\% \end{array}$$
where $M_{s}$ is the saturation magnetization, and $M_{r,\;z-axis}$ is the remanent magnetism in the *z*-axis direction. For the none easy-plane FeSi_3.5_ composite, the hysteresis loops of the in-plane and out-of-plane direction coincide. Neither the in-plane direction or out-of-plane direction was magnetized to saturation (about 154 emu/g). For the easy-plane FeSi_3.5_ composite, the hysteresis loops of the in-plane and out-of-plane direction are different. The in-plane direction is more easily magnetized to saturation (about 173.5 emu/g) than the out-of-plane direction (Table 1).

### 3.3. Complex Permeability and Magnetic Spectra Simulation

Figure 3 shows the magnetic spectra and spectra simulation of the none easy-plane FeSi_3.5_ and the easy-plane FeSi_3.5_ composite with 60 vol%. For the none easy-plane FeSi_3.5_ composite, the real part of permeability is 30. It has a resonance peak at 20 MHz in the imaginary part of permeability. For the easy-plane FeSi_3.5_ composite, the real part of permeability is 62. Two resonance peaks appear in the imaginary part of the permeability. The domain wall resonance peak is observed in a lower frequency (100 MHz), and the natural resonance is observed in a higher frequency (1.5 GHz). According to the domain wall motion mechanism and spin rotation mechanism, magnetic spectra can be simulated by Equations (3) and (4) [24,25,26,27,28]:(3)$$\begin{array}{cc} \mu \prime & =\mu_{dw}^{\prime }+\mu_{spin}^{\prime }\\ & =\frac{\omega_{dw}^{2}\chi_{dw}\left(\omega_{dw}^{2}-\omega^{2}\right)}{{\left(\omega_{dw}^{2}-\omega^{2}\right)}^{2}+\omega^{2}\beta^{2}}+\frac{\chi_{spin}\omega_{spin}^{2}\left[\left(\omega_{spin}^{2}-\omega^{2}\right)+\omega^{2}\alpha^{2}\right]}{{\left[\omega_{spin}^{2}-\omega^{2}\left(1+\alpha^{2}\right)\right]}^{2}+4\omega^{2}\omega_{spin}^{2}\alpha^{2}}+1 \end{array}$$
(4)$$\begin{array}{cc} \mu^{\prime \prime } & =\mu_{dw}^{\prime \prime }+\mu_{spin}^{\prime \prime }\\ & =\frac{\omega_{dw}^{2}\chi_{dw}\omega \beta }{{\left(\omega_{dw}^{2}-\omega^{2}\right)}^{2}+\omega^{2}\beta^{2}}+\frac{\chi_{spin}\omega_{spin}^{2}\omega \alpha \left[\omega_{spin}^{2}+\omega^{2}\left(1+\alpha^{2}\right)\right]}{{\left[\omega_{spin}^{2}-\omega^{2}\left(1+\alpha^{2}\right)\right]}^{2}+4\omega^{2}\omega_{spin}^{2}\alpha^{2}} \end{array}$$
where $\mu_{dw}^{\prime }$ and $\mu_{spin}^{\prime }$ are the real part of permeability for the domain wall motion mechanism and spin rotation mechanism, and $\mu_{dw}^{\prime \prime }$ and $\mu_{spin}^{\prime \prime }$ are the imaginary part of permeability for the domain wall motion mechanism and spin rotation mechanism, respectively. $\omega_{dw}$, $\chi_{dw}$ and β are the domain wall resonance frequency, static susceptibility and damping factor for the domain wall component. $\omega_{spin}$, $\chi_{spin}$ and α is the resonance frequency, static susceptibility and damping factor for the spin rotation component. ω is the frequency of the applied field (ω=2πf). According to Equations (3) and (4), the magnetic spectra are simulated and six parameters ($\omega_{dw}$, $\chi_{dw}$, β, $\omega_{spin}$, $\chi_{spin}$, α) are obtained (Figure 3b,c and Table 2). 

The theoretical domain wall resonance frequency and natural resonance frequency of the none easy-plane FeSi_3.5_ composite are 120 MHz and 350 MHz. The real part of permeability for the domain wall component and spin rotation are 13 and 14. The theoretical domain wall resonance peak and natural resonance peak of the easy-plane FeSi_3.5_ composite are 840 MHz and 7 GHz. The real part of permeability for the domain wall component and spin rotation are 45 and 20. 

### 3.4. Power Loss and Loss Separation

Figure 4 shows the power loss and loss separation of the none easy-plane FeSi_3.5_ composite and the easy-plane FeSi_3.5_ composite. The easy-plane FeSi_3.5_ composite exhibits a desired total loss $P_{cv}$ of 500.42 kW/m^3^ under the test conditions of 8 mT and 3 MHz. In general, the $P_{cv}$ rapidly increases with increasing frequency, and the maximum $P_{cv}$ of the easy-plane FeSi_3.5_ composite (500.42 kW/m^3^) is much lower than that of the none easy-plane FeSi_3.5_ composite (2785.8 kW/m^3^).

According to classic Bertottit’s loss separation theory, the total loss of $P_{cv}$ can be separated into three different parts: hysteresis loss $P_{hyst}$, eddy current loss $P_{eddy}$ and excess loss $P_{exc}$. The expression can be described [29]:(5)$$\begin{array}{c} P_{cv}=c_{hyst}B_{m}^{\alpha }f+c_{eddy}B_{m}^{2}f^{2}+c_{exc}B_{m}^{x}f^{y} \end{array}$$
where $c_{hyst}$ and α are the coefficients of hysteresis loss, $c_{eddy}$ is the coefficient of current eddy loss, and $c_{exc}$, x and y are the coefficients of excess loss. The coefficients in each term are shown in Table 3.

For the none easy-plane FeSi_3.5_ composite, $P_{exc}$ is the largest proportion of the total loss, about 2435.2 kW/m^3^ (8 mT and 3 MHz), and $P_{hyst}$ and $P_{eddy}$ are small percentages of the total loss, about 224.79 kW/m^3^ and 125.77 kW/m^3^. For the easy-plane FeSi_3.5_ composite, $P_{eddy}$ is the smallest proportion of the total loss, about 0.689 kW/m^3^, and $P_{hyst}$ and $P_{exc}$ are main part of total loss, about 294.8 kW/m^3^ and 204.93 kW/m^3^. Although the $P_{hyst}$ of the none easy-plane FeSi_3.5_ composite is less than that of easy-plane FeSi_3.5_ composite, the difference between them is only 70 kW/m^3^ under the test conditions of 8 mT and 3 MHz. For the $P_{eddy}$ and $P_{exc}$, the advantages of the easy-plane FeSi_3.5_ composite are shown. The $P_{eddy}$ of the easy-plane FeSi_3.5_ composite is $\frac{1}{200}$ for that of the none easy-plane FeSi_3.5_ composite. The $P_{exc}$ of the easy-plane FeSi_3.5_ composite is $\frac{1}{20}$ for that of the none easy-plane FeSi_3.5_ composite.

## 4. Discussion

The easy-plane FeSi_3.5_ composite has excellent magnetic properties and loss properties at higher frequencies compared with the none easy-plane FeSi_3.5_ composite. The easy-plane FeSi_3.5_ composite has higher permeability and higher ferromagnetic resonance frequencies than the none easy-plane FeSi_3.5_ composite, which is beneficial for reducing the excitation current, turns and achieving device miniaturization. And the $P_{cv}$ of the easy-plane FeSi_3.5_ composite (500.42 kW/m^3^) is lower and about $\frac{1}{5}$ for than that of the none easy-plane FeSi_3.5_ composite (2785.8 kW/m^3^). It is emphasized that the easy-plane FeSi_3.5_ composite has lower $P_{cv}$ compared with the none easy-plane FeSi_3.5_ composite because of the significant reduction in $P_{exc}$ rather than the decrease of $P_{eddy}$. The $P_{eddy}$ of the none easy-plane FeSi_3.5_ composite and easy-plane FeSi_3.5_ composite is only a small proportion of total loss (as shown in Figure 4b,c). The $P_{eddy}$ of the none easy-plane FeSi_3.5_ composite and easy-plane FeSi_3.5_ composite, respectively, only accounts for 4.5% and 0.14% of the total loss at 8 mT and 3 MHz (as shown in Figure 5b,e). Both the none easy-plane FeSi_3.5_ composite and easy-plane FeSi_3.5_ composite have a particle size of microns, and are composite materials (high resistivity). Therefore, they have a low intra-particle eddy current loss $P_{eddy}^{intra}$ and inter-particle eddy current loss $P_{eddy}^{inter}$. And the easy-plane FeSi_3.5_ composite with smaller particle thickness can have a lower $P_{eddy}$ (as shown in Figure 4e) compared with the none the easy-plane FeSi_3.5_ composite. Therefore, the $P_{eddy}$ will not become a relatively large part of the loss and a major issue at higher frequencies of SMCs. 

It is found that $P_{exc}$ becomes the main loss when increasing the frequency, according to the experimental results (as shown in Figure 4b,c). The $P_{exc}$ of none the easy-plane FeSi_3.5_ composite and easy-plane composite both occupies the larger portion of the total loss, which accounts for 87.4% (2435.2 kW/m^3^:2785.8 kW/m^3^) and 40.9% (204.93 kW/m^3^: 500.42 kW/m^3^) of the total loss at 8 mT and 3 MHz, respectively (as shown in Figure 5b,e). Whether the none easy-plane FeSi_3.5_ composite or easy-plane FeSi_3.5_ composite, the $P_{exc}$ both dramatically increases with frequency. For the none easy-plane FeSi_3.5_ composite, the $P_{exc}$ sharply increases and becomes the main part of the total loss in the range of 200 kHz to 1200 kHz, accounting for about 80% of the total loss. For the easy-plane FeSi_3.5_ composite, the main loss is $P_{hyst}$ at a low frequency. As the frequency increases, the $P_{exc}$ of the easy-plane FeSi_3.5_ composite rapidly increases and becomes one of the main losses. Therefore, as the frequency increases, the dramatical increase of $P_{exc}$ becomes a serious problem to soft magnetic materials in higher frequency applications, especially after reaching the MHz band.

Both of the none easy-plane FeSi_3.5_ composite and easy-plane FeSi_3.5_ composite can maintain a low loss at a lower frequency, but the easy-plane FeSi_3.5_ composite has an obvious advantage in the high-frequency band. For the none easy-plane FeSi_3.5_ composite, the dramatical increase in $P_{exc}$ alongside frequency leads to a rapid increase in total loss, the $P_{exc}$ is over $P_{hyst}$ at 250 kHz, about 20 kW/m^3^; along with the rapid increase in frequency, the $P_{exc}$ accounts for about 80% of the total loss at 1 MHz, about 342.8 kW/m^3^; the $P_{exc}$ reaches about 2435.2 kW/m^3^ while the frequency continuously increases to 3 MHz (Figure 5c). For the easy-plane FeSi_3.5_ composite, the $P_{exc}$ is about 0.75 kW/m^3^ at 250 kHz, 24.61 kW/m^3^ at 1 MHz and 204.93 kW/m^3^ at 3 MHz (Figure 5f). Compared with the none easy-plane FeSi_3.5_ composite, the $P_{exc}$ in the easy-plane FeSi_3.5_ composite is lower. This phenomenon is possibly related to the domain wall resonance frequency, and we will continue to study this in the future. As shown in Figure 3, the domain wall resonance frequency of the easy-plane FeSi_3.5_ composite and none easy-plane FeSi_3.5_ composite is 100 MHz and 9 MHz, respectively. Due to the domain wall resonance frequency of the none easy-plane FeSi_3.5_ composite being low, the $P_{exc}$ dramatically increases with frequency and the frequency is only 250 kHz when the $P_{exc}$ exceeds the $P_{hyst}$ (Figure 5c). For the easy-plane FeSi_3.5_ composite, $P_{exc}$ still increases with frequency, but the rate of the increase of $P_{exc}$ with frequency is lower, due to the high-domain-wall frequency, and $P_{exc}$ exceeds the $P_{hyst}$ only when the frequency is higher than 3 MHz (Figure 5f). The high-domain-wall resonance frequency can push the frequency point, where the excess loss dramatically increases to a higher frequency, thus the easy-plane material with a higher domain wall resonance frequency can well inhibit the increase in excess loss with frequency.

The performance factor (PF) is a parameter that describes the energy transformation efficiency and can be expressed as the product of $B_{m}$ and f by Equation (1):(6)$$\begin{array}{c} PF=B_{m}f \end{array}$$

The PF is calculated by simulated parameters in Table 2. It is found that PF and total loss of the easy-plane FeSi_3.5_ composite is much more excellent than that of the none easy-plane FeSi_3.5_ composite. This indicates that the easy-plane FeSi_3.5_ composite has a higher transformation efficiency and lower loss compared with the none easy-plane FeSi_3.5_ composite, and that the easy-plane FeSi_3.5_ composite is more suitable for operating at higher frequencies. In addition, the currently published results of SMCs operating at higher frequencies are compared with those of the easy-plane FeSi_3.5_ composite. The easy-plane FeSi_3.5_ composite still has an advantage in permeability, operating frequency and transformation efficiency (as shown Figure 6b in Table 4).

## 5. Conclusions

The easy-plane FeSi_3.5_ composite has an ultra-low loss at 10 MHz, 4mT (372.88 kW/m^3^). The reason for the ultra-low loss of the easy-plane FeSi_3.5_ composite at higher frequencies is because of setting up a control group (the none easy-plane FeSi_3.5_ composite). For the first time, we found that the root cause of the failure in soft magnetic composites to work properly at higher frequencies was due to the dramatic increase in the $P_{exc}$. The rate of increase and the percentage of the $P_{exc}$ is much larger than that of the $P_{hyst}$ and the $P_{eddy}$. The $P_{exc}$ accounts for a large portion of the $P_{cv}$; the portion continues to sharply increase with frequency, and conventional loss reduction methods have no effect on the $P_{exc}$. In addition, we also found that the easy-plane composite can effectively reduce the $P_{exc}$. Compared with the none easy-plane composite, the easy-plane composite can greatly reduce $P_{exc}$ from 2435.2 kW/m^3^ to 204.93 kW/m^3^. This substantial reduction can greatly reduce the $P_{cv}$. And the easy-plane composite also has excellent magnetic properties, high permeability and ferromagnetic resonance frequencies. Thus, the easy-plane composite has strong potential to be applied for inductor and transformer cores at higher frequencies, especially at the MHz band and above.

## Figures and Tables

**Figure 1 materials-16-05133-f001:**
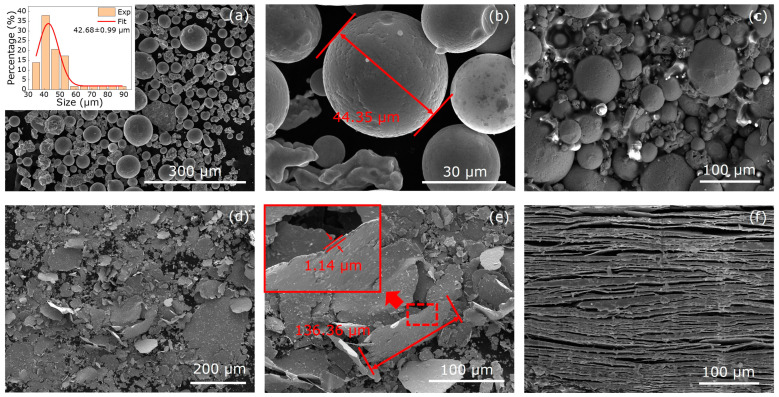
The SEM of the none easy-plane FeSi_3.5_ particles (**a**,**b**) and the cross sections of the composite (**c**), the SEM of the easy-plane FeSi_3.5_ particles (**d**,**e**) and the cross sections of the composite (**f**).

**Figure 2 materials-16-05133-f002:**
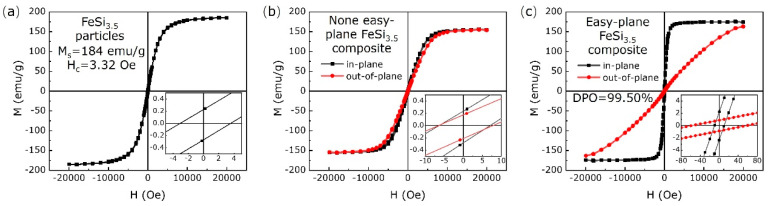
The hysteresis loop of FeSi_3.5_ particles (**a**), hysteresis loops and *DPO* of the none easy-plane FeSi_3.5_ composite (**b**) and easy-plane FeSi_3.5_ composite (**c**).

**Figure 3 materials-16-05133-f003:**
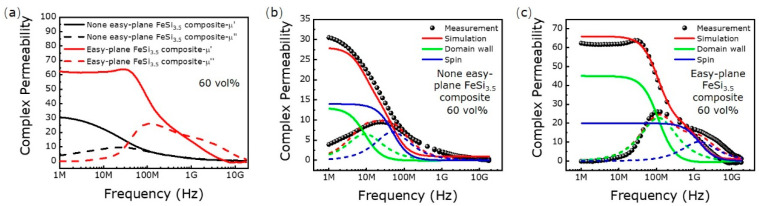
The complex permeability (**a**) and the spectra simulation of the none easy-plane FeSi_3.5_ composite (**b**) and the easy-plane FeSi_3.5_ composite (**c**) with 60 vol%.

**Figure 4 materials-16-05133-f004:**
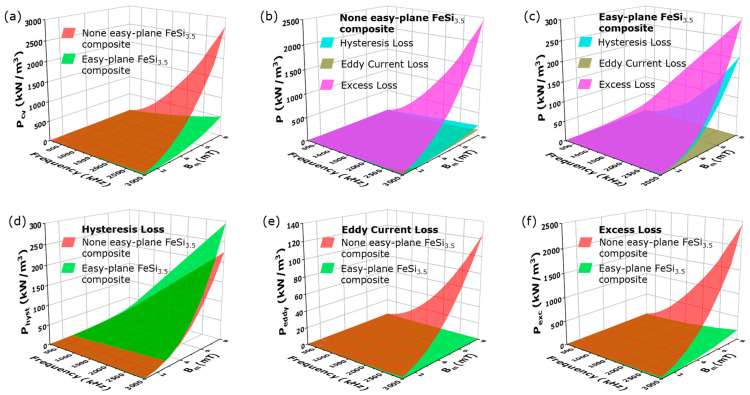
The total loss (**a**), loss separation (**b**,**c**), hysteresis loss (**d**), eddy current loss (**e**) and excess loss (**f**) of the none easy-plane FeSi_3.5_ composite and easy-plane FeSi_3.5_ composite.

**Figure 5 materials-16-05133-f005:**
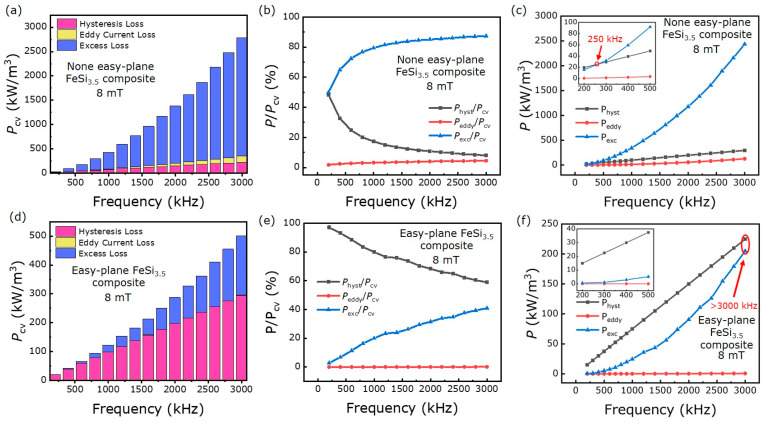
The percentage of loss of each part and loss separation of the none easy-plane FeSi_3.5_ composite (**a**–**c**) and easy-plane FeSi_3.5_ composite (**d**–**f**).

**Figure 6 materials-16-05133-f006:**
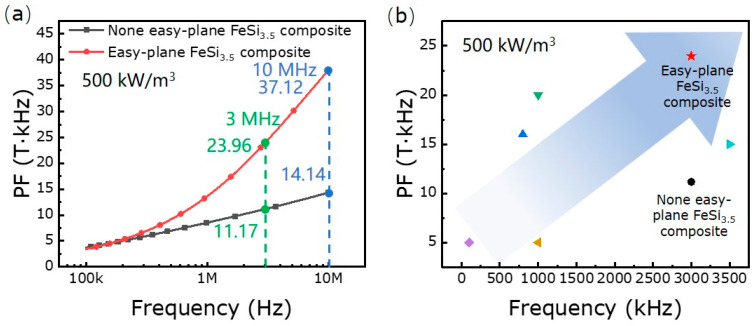
The PF calculate curve of the none easy-plane FeSi_3.5_ composite and the easy-plane FeSi_3.5_ composite at 500 kW/m^3^ (**a**). The PF of different SMCs at 500 kW/m^3^ (**b**) [19,20,21,22].

**Table 1 materials-16-05133-t001:** The $M_{s}$, $M_{r}$ and $H_{c}$ of the in-plane and out-of-plane direction and *DPO* of the none easy-plane FeSi_3.5_ composite and easy-plane FeSi_3.5_ composite.

	In-Plane			Out-of-Plane			*DPO*
	$M_{s}$	$M_{r}$	$H_{c}$	$M_{s}$	$M_{r}$	$H_{c}$	
None easy-plane FeSi_3.5_	154.45	0.24	6.71	153.78	0.18	7.18	
Easy-Plane FeSi_3.5_	173.53	2.28	10.19	163.00	0.87	57.79	99.50%

**Table 2 materials-16-05133-t002:** The six parameters in the magnetic spectra of the none easy-plane FeSi_3.5_ composite and easy-plane FeSi_3.5_ composite.

	$\mu_{i}$		Domain Wall Motion				Spin Rotation		
		$\chi_{dw}$	$f_{dw}\;\left(MHz\right)$	$f_{dw}^{\mu \prime \prime \;\left(max\right)}\;\left(MHz\right)\;$	*β*	$\chi_{spin}$	$f_{spin}\;\left(GHz\right)$	$f_{spin}^{\mu \prime \prime \;\left(max\right)}\;\left(GHz\right)\;$	*α*
None easy-plane FeSi_3.5_	28	13	120	9	1 × 10^10^	14	0.35	0.05	7
Easy-Plane FeSi_3.5_	66	45	840	100	4 × 10^10^	20	7	1.5	5

**Table 3 materials-16-05133-t003:** Simulated parameters for loss separation of the none easy-plane FeSi_3.5_ composite and easy-plane FeSi_3.5_ composite.

	$P_{hyst}$		$P_{eddy}$			$P_{exc}$	
	$c_{hyst}$	*α*	$c_{eddy}^{inter}$	$c_{eddy}^{intra}$	$c_{exc}$	*x*	*y*
None easy-plane FeSi_3.5_ composite	9.2463	2.43	5.14 × 10^−12^	2.18 × 10^−4^	4.02 × 10^−4^	2.21	1.76
Easy-Plane FeSi_3.5_ composite	42.897	2.69	8.18 × 10^−7^	3.78 × 10^−7^	4.16 × 10^−6^	2.43	1.97

**Table 4 materials-16-05133-t004:** Magnetic properties and loss properties of various up-to-date published SMCs above 1 MHz.

SMCs	Initial Permeability	Power Loss	PF (under 500 kW/m^3^)	
The easy-plane FeSi_3.5_ composite	62.5	500.42 kW/m^3^ (3 MHz, 8 mT)	23.96	This work
Amorphous FeSiCr + carbonyl iron @ resin	37	560 kW/m^3^ (1 MHz, 20 mT)	16	[19]
Amorphous Fe_73_Si_6_B_10_P_5_C_3_Mo_3_ @ epoxy resin	34.63	~500 kW/m^3^ (1 MHz, 20 mT)	20	[21]
Amorphous FeSiCr/phosphating process and polyimide coating	47.5	7086 kW/m^3^ (1 MHz, 50 mT)	5	[20]
Flake pure iron @ binder composite	67 (at 1 MHz)	306.6 kW/m^3^ (1 MHz, 3 mT)	~5	[22]
	10 (at 3.5 MHz)	3.285 kW/m^3^ (3.5 MHz, 0.1 mT)	~15	

## Data Availability

Not applicable.

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
