# Peer review of "A Study on an Easy-Plane FeSi3.5 Composite with High Permeability and Ultra-Low Loss at the MHz Frequency Band"

_materials, 2023, doi:10.3390/ma16145133_

Round 1
Reviewer 1 Report
The authors compare the magnetic properties of composites with none-easy-plane and easy-plane FeSi particles. The content of particles in both cases is the same and amounts to 60 vol%. Main conclusion: easy-plane composite is better than none-easy-plane.
However, the reviewer wants to point out that the none-easy-plane particles are spherical, while the easy-plane particles are platelets. So, the shape and size of none-easy-plane and easy-plane particles differ significantly. Probably, the magnetic states of the particles and the interactions between them are also significantly different. Therefore, the main question arises: how correct is the comparison of the composites presented in the article and how general are the conclusions obtained? Only one thing can be said based on the results presented: one easy-plane composite is better than one none-easy-plane composite. Perhaps the authors should consider composites with particles of the same shape but different sizes, as well as composites with different particle content. It would also be interesting to compare samples with different DPO.
Authors also need to pay attention to the following.
- Pexc (line 71) and Pcv (line 72) are not described at the first mention.
- A detailed description of the process for obtaining none-easy-plane and easy-plane particles is necessary. It is recommended to present the particle size distribution and the results of the study of the structural state.
- What is the strength of the rotating magnetic field? (line 85)
- What is the thickness of the ring-shaped composite? (line 87)
- For which particles (none-easy-plane or easy-plane) is the loop shown in Figure 2a?
Author Response
Dear Editors:
Thank you very much for your reply. For your valuable comments, we have considered the certification, given a reasonable explanation and made some modifications (yellow part). If there are still problems, please contact us as soon as possible and we will respond as soon as possible.
Here we attach review comments and answer them as follows:
For Reviewer 1:
However, the reviewer wants to point out that the none-easy-plane particles are spherical, while the easy-plane particles are platelets. So, the shape and size of none-easy-plane and easy-plane particles differ significantly. Probably, the magnetic states of the particles and the interactions between them are also significantly different. Therefore, the main question arises: how correct is the comparison of the composites presented in the article and how general are the conclusions obtained? Only one thing can be said based on the results presented: one easy-plane composite is better than one none-easy-plane composite. Perhaps the authors should consider composites with particles of the same shape but different sizes, as well as composites with different particle content. It would also be interesting to compare samples with different DPO.
Answer: Thank you for your suggestions. In terms of microscopic morphology, it is true that the none easy-plane FeSi3.5 particles are spherical and the easy-plane FeSi3.5 particles are flaky. However, the “easy-plane” presented in this paper is a physical concept, the explanation of which has been mentioned in the introduction. What is discussed in this paper is the effect of this physical mechanism on the power loss of MHz. In our current study, this phenomenon, which can greatly reduce power loss at MHz band, is not only present the easy-plane FeSi3.5 composite, but also in the easy-plane FeNi composite [1], the easy-plane α-Fe composite (verified). For the different particle size, different particle content and different DPO are good research directions, and we are currently doing related research.
[1] G. Wang, J. Zhang, Z. Zheng, L. Qiao, T. Wang, F. Li, Comparative study on the high frequency performances of the easy-plane FeNi@SiO2 powder soft magnetic composite, Current Applied Physics 41 (2022) 73-80.
- Pexc (line 71) and Pcv (line 72) are not described at the first mention.
Answer: The manuscript has been revised. Thank you for your suggestions.
- A detailed description of the process for obtaining none-easy-plane and easy-plane particles is necessary. It is recommended to present the particle size distribution and the results of the study of the structural state.
Answer: In this paper, the easy-plane FeSi3.5 particles were obtained from the none easy-plane FeSi3.5 particles after easy-plane processing. The easy-plane processing will not be described due to the need for confidentiality. The particle size distribution of non-faceted particles has been added in the manuscript.
- What is the strength of the rotating magnetic field? (line 85)
Answer: The manuscript has been revised. Thank you for your suggestions.
- What is the thickness of the ring-shaped composite? (line 87)
Answer: The manuscript has been revised. Thank you for your suggestions.
- For which particles (none-easy-plane or easy-plane) is the loop shown in Figure 2a?
Answer: The particles here are the original particles of FeSi3.5, which can also be taken as the none easy-plane FeSi3.5 particles. However, it should be noted that the VSM results of the original powder are provided only to show the intrinsic properties of the FeSi3.5 original particles, and Fig 1. (b) and (c) is only to illustrate the difference between easy-plane FeSi3.5 composite and none easy-plane FeSi3.5 composite.

Reviewer 2 Report
In general, despite the rather topical topic of the study, the presented work requires significant revision both in terms of reflecting the results and in technical terms.
1. The main remark to this article is that in the abstract, the authors do not disclose the composition of the composite under study, using the rather rare term “easy-plane composite” for description, thus not letting the readers of this article understand what specific composition of the composite is in question. Moreover, in the title they give the composition of this composite, but in the abstract there is no description of how exactly it was obtained.
2. The described problems with the occurrence of the effect of overheating at high frequencies should be described in more detail, since this is a very important problem in the application of magnetic materials.
3. The presented SEM images should be shown in more detail with the reflection of the dimensions between the layers, if possible. It is also not entirely clear the need to bring spherical particles in Figure 1.
4. The authors say that they were able to obtain a unique easy-plane composite, but do not provide any data on its phase and elemental composition, as well as structural features.
5. Have the authors established relationships between changes in magnetic properties and the density or structural features of the resulting easy-plane composite?
6. For technical comments, the authors should correct all captions and bring them in the same style.
7. Also, during the final layout, you should once again proofread the text and correct minor flaws and typos.
Author Response
Dear Editors:
Thank you very much for your reply. For your valuable comments, we have considered the certification, given a reasonable explanation and made some modifications (yellow part). If there are still problems, please contact us as soon as possible and we will respond as soon as possible.
Here we attach review comments and answer them as follows:
For Reviewer 2:
- The main remark to this article is that in the abstract, the authors do not disclose the composition of the composite under study, using the rather rare term “easy-plane composite” for description, thus not letting the readers of this article understand what specific composition of the composite is in question. Moreover, in the title they give the composition of this composite, but in the abstract there is no description of how exactly it was obtained.
Answer: The abstract has been revised. However, it should be noted that this paper is intended to propose a new method capable of reducing loss (the easy-plane FeSi3.5 composite). Not only in the easy-plane FeSi3.5 composite, but also in the easy-plane FeNi composite [1], the easy-plane α-Fe composite (verified) can also have the same effect.
[1] G. Wang, J. Zhang, Z. Zheng, L. Qiao, T. Wang, F. Li, Comparative study on the high frequency performances of the easy-plane FeNi@SiO2 powder soft magnetic composite, Current Applied Physics 41 (2022) 73-80.
- The described problems with the occurrence of the effect of overheating at high frequencies should be described in more detail, since this is a very important problem in the application of magnetic materials.
Answer: The effect of overheating, which affects the lifetime of the device and increases the loss of electrical energy, is currently treated as a drawback. This paper finds that the power loss of magnetic materials at high frequency is very huge, so the power loss results in the heating of magnetic materials, and if the frequency continues to increase, the heating will increase and will affect the work of the whole device. At present, there is less research on this overheating effect, when done as a phenomenon in the background section for discussion. Therefore, we have introduced some articles to better illustrate the effect of overheating at high frequencies.
- The presented SEM images should be shown in more detail with the reflection of the dimensions between the layers, if possible. It is also not entirely clear the need to bring spherical particles in Figure 1.
Answer: SEM is mainly to show the cross-sectional structure of the easy-plane FeSi3.5 composite and none easy-plane FeSi3.5 composite. The main purpose is to further prove the “easy-plane” physical properties.
- The authors say that they were able to obtain a unique easy-plane composite, but do not provide any data on its phase and elemental composition, as well as structural features.
Answer: The easy-plane FeSi3.5 composite and none easy-plane FeSi3.5 composite mentioned in this paper are named according to the physical mechanism, the easy-plane FeSi3.5 composite is from none easy-plane FeSi3.5 composite after facile treatment to the two, in addition to the physical mechanism and morphological differences, there is no difference in composition and crystal structure (shown in Fig. 1).
Fig.1 The XRD of easy-plane FeSi3.5 composite and none easy-plane FeSi3.5 composite
- Have the authors established relationships between changes in magnetic properties and the density or structural features of the resulting easy-plane composite?
Answer: The easy-plane FeSi3.5 composite has higher permeability and ferromagnetic resonance frequencies compared with the none easy-plane FeSi3.5 composite. This is due to the physical properties of “easy-plane”.
- For technical comments, the authors should correct all captions and bring them in the same style.
Answer: The manuscript has been revised. Thank you for your suggestions.
- Also, during the final layout, you should once again proofread the text and correct minor flaws and typos.
Answer: The manuscript has been revised. Thank you for your suggestions.

Reviewer 3 Report
An easy-plane composite is proposed justifying the real reason why magnetic materials do not work properly due to overheat. SEM photos of the cross sections of various composites are provided and the magnetization curves in-plane and out-of-plane are given. The complex permittivities are evaluated across the operational frequency axis for various composites while the various losses are computed also as functions of frequency. A map with the performance of various designs is finally provided.
The paper deals with an interesting topic, it is well-written and seems physically sound but extensive improving modifications are required in order to become publishable at Materials. In particular:
(A) Where is the novelty? A more extensive novelty statement is required in the revised version.
(B) The authors should discuss extensively the application that these materials can have in building memory and multistability elements [1,2].
(C) Some competing materials that combine high permeability with low losses are missing from the comparison [3].
[1] Angular Memory of Photonic Metasurfaces, IEEE Transactions on Antennas and Propagation, 2021.
[2] Multistability in coupled nonlinear metasurfaces, IEEE Transactions on Antennas and Propagation, 2022.
[3] Modeling the Hysteresis Loop of Ultra-High Permeability Amorphous Alloy for Space Applications, Materials, 2018.
Author Response
Dear Editors:
Thank you very much for your reply. For your valuable comments, we have considered the certification, given a reasonable explanation and made some modifications (yellow part). If there are still problems, please contact us as soon as possible and we will respond as soon as possible.
Here we attach review comments and answer them as follows:
For Reviewer 3:
- Where is the novelty? A more extensive novelty statement is required in the revised version.
Answer: We modified the manuscript and we added the power loss results at 10 MHz. In the current power loss studies of SMC, the results with ultra-low loss at 10MHz are rare. Our experimental comparison shows that SMC with the physical property of “easy-plane” can greatly reduce the power loss. This unique magnetic structure can reduce the loss is one of our innovation points. It is also one of our innovations to prove that the excess loss is the main cause for high loss in MHz band.
- The authors should discuss extensively the application that these materials can have in building memory and multistability elements.
Answer: Thank you very much for the suggestion, building memory and multistability elements are two good application scenarios. But currently the main application scenarios for SMC are transformers, inductors, etc. Especially these applications are affected by the third-generation wide bandgap semiconductors (WBG) and need to reach higher operating frequencies. This paper also addresses this problem by proposing the easy-plane FeSi3.5 composite to solve the problem of large high-frequency power loss.
- Some competing materials that combine high permeability with low losses are missing from the comparison.
Answer: Thank you very much for your suggestion, this high magnetic permeability result is indeed worth quoting and we have modified it (see citation 4). Although there are more power loss studies of SMC, these studies are in the kHz range. This paper focuses on the power loss studies above MHz, and some of the loss studies in the MHz range have been listed in Table. 4 in this paper.

Round 2
Reviewer 1 Report
Unfortunately, the reviewer cannot change his decision. Despite serious comments, the article was not properly revised.
It is necessary to note the following fundamental point. Can another scientist get the same results based on the description of the experiment in this article?
Authors need to pay attention to the description of the object of study. The question of how the none-easy-plane and easy-plane particles are obtained remains open.
What is the size distribution of easy-plane particles? Why does an easy plane appear? Is it connected with the production technology, with the conditions for the subsequent processing of the composite or with the shape of the composite?
Section 3.3 does not discuss the role of the shape and size of particles, as well as the distance between them.
Moreover, the authors did not work on the previous comment of the reviewer: The shape and size of none-easy-plane and easy-plane particles differ significantly. Probably, the magnetic states of the particles and the interactions between them are also significantly different. Therefore, the main question arises: how correct is the comparison of the composites presented in the article and how general are the conclusions obtained? Only one thing can be said based on the results presented: one easy-plane composite is better than one none-easy-plane composite.
Reviewer 2 Report
The authors answered all the questions posed, the article can be accepted for publication in an updated form.
Author Response
Thank you!
Round 3
Reviewer 1 Report
The authors tried to respond to the comments of the reviewer. Nevertheless, the reviewer is surprised by the desire of the authors to hide some details of the experiment. Such a strategy is suitable for confidential commercial or government reports, but it is not suitable for open printing. The reviewer wants to hope that the readers of the journal, if necessary, will be able to repeat the results obtained by the authors.